ecology, neuroscience, behaviour

zebra stripes, optic flow, tabanid, vision, insect flight

**Author for correspondence:**
Martin J. How
e-mail: m.how@bristol.ac.uk

# Zebra stripes, tabanid biting flies and the aperture effect

Martin J. How[1], Dunia Gonzales[1], Alison Irwin[1] and Tim Caro[1,2]

[1]School of Biological Sciences, University of Bristol, 24 Tyndall Avenue, Bristol BS8 1TQ, UK
[2]Center for Population Biology, University of California, Davis, CA 95616, USA

 MJH, 0000-0001-5135-8828; TC, 0000-0001-6804-8519

Of all hypotheses advanced for why zebras have stripes, avoidance of biting fly attack receives by far the most support, yet the mechanisms by which stripes thwart landings are not yet understood. A logical and popular hypothesis is that stripes interfere with optic flow patterns needed by flying insects to execute controlled landings. This could occur through disrupting the radial symmetry of optic flow via the aperture effect (i.e. generation of false motion cues by straight edges), or through spatio-temporal aliasing (i.e. misregistration of repeated features) of evenly spaced stripes. By recording and reconstructing tabanid fly behaviour around horses wearing differently patterned rugs, we could tease out these hypotheses using realistic target stimuli. We found that flies avoided landing on, flew faster near, and did not approach as close to striped and checked rugs compared to grey. Our observations that flies avoided checked patterns in a similar way to stripes refutes the hypothesis that stripes disrupt optic flow via the aperture effect, which critically demands parallel striped patterns. Our data narrow the menu of fly-equid visual interactions that form the basis for the extraordinary colouration of zebras.

## 1. Introduction

The unusual and striking colouration pattern of the three species of extant zebra (*Equus* sp.) has generated many intriguing functional explanations over the last 150 years [1]. For convenience, they, and the evidence for and against them, can be collapsed into four principal themes. (i) Early ideas about stripes being a form of crypsis against predators have now been dismissed on grounds that large carnivores are only able to resolve stripes very close up [2] and by experiments showing that zebra stripes are highly conspicuous to human observers [3]. (ii) Conjectures about stripes confusing predators are poorly supported by observations of plains zebras (*Equus quagga*) fleeing in ways that do not enhance protean behaviour, nor obscure the outline of individual animals, and because they do not promote motion dazzle or cause lions to misdirect their attack [3]. Most damning, lions kill zebras more than expected from their abundance across 40 study sites in Africa [4] suggesting that confusion is an unlikely functional explanation for stripes. (iii) A third theme, that black and white stripes have different heat signatures [5] that set up convection currents that cool zebras, could only operate under very restricted circumstances: over the animal's dorsum, not over its flank or legs, and, problematically, not under breezy conditions or when the animal moves, as these would negate convection currents acting anywhere on the body [3]. Moreover, experimental tests reveal no cooling benefits associated with striped objects or striped pelts [6]. (iv) In contrast with these ideas, stripes are an established potent force in reducing landings of biting flies based on experimental studies with striped artificial targets [7–9], horse models [10], human models [11], painted cows [12], and comparisons of live plains zebras with domestic horses [13].

Recently, we showed that tabanid horseflies approach domestic horses wearing striped rugs as often as the same horses covered with black or white

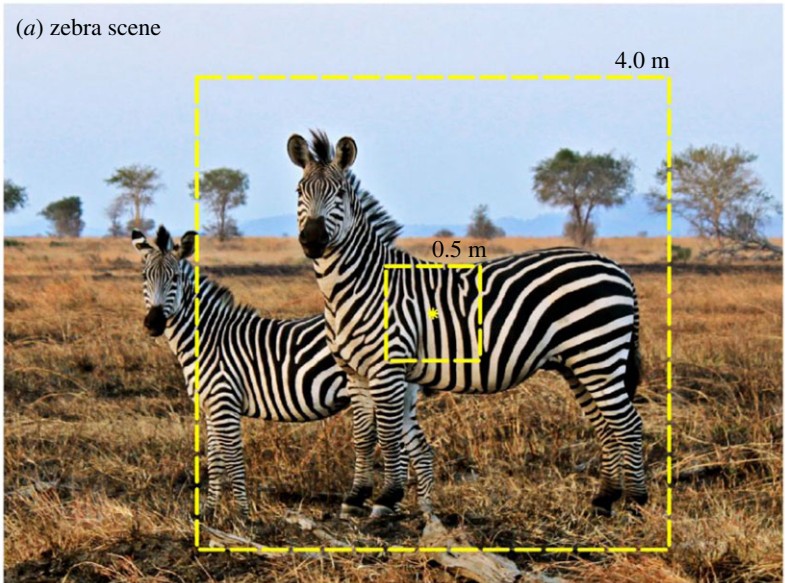

(a) zebra scene

4.0 m

0.5 m

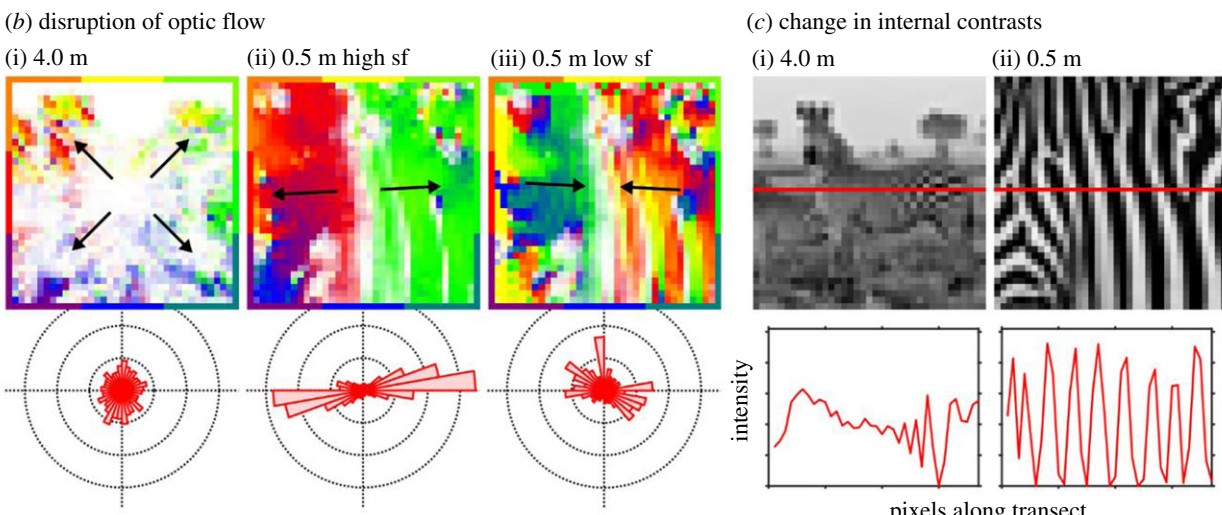

(b) disruption of optic flow

(i) 4.0 m          (ii) 0.5 m high sf          (iii) 0.5 m low sf

(c) change in internal contrasts

(i) 4.0 m          (ii) 0.5 m

intensity

pixels along transect

**Figure 1.** Illustration of the main test hypotheses. (a) Zebra scene (Wikimedia Commons). Dashed squares illustrate horsefly views chosen at 4.0 m and 0.5 m distances. Yellow star illustrates a fictive landing site towards which a virtual fly is heading. (b) Motion maps generated from the same down-sampled images expanding from the central landing point. Colour encodes motion direction (according to the surrounding border scale) and saturation encodes motion strength at (i) 4.0 m illustrating typical optic flow, (ii) 0.5 m with high spatial sample frequency (sf) illustrating the aperture effect, and (iii) at 0.5 m with low spatial sample frequency illustrating spatio-temporal aliasing. The distribution of motion vectors across each of these scenes is represented in the polar histograms below. (c) Scene at the two viewing distances down-sampled to approximate tabanid visual resolution (0.5 cycles/°). Horizontal red line indicates an image transect, the intensity values of which are plotted below.

rugs but that far fewer tabanids actually landed on horses with these striped external appearances. Parallel observations revealed that tabanids failed to decelerate when approaching zebras compared to horses and, anecdotally, flies were seen to bump into zebra pelage or fly past it [13].

The mechanism by which stripes exert these effects remains opaque. Given the striking appearance of zebras (figure 1a), it is often argued that regularly spaced stripes of zebra pelage somehow confuse the visual system of flying insects [10,13,14]. Specifically, stripes could interfere with the control of visually guided flight by disrupting optic flow [14]. Optic flow is the pattern of apparent motion caused by relative movement between an observer and the scene. As a fly, moving at a constant speed, approaches an object, the surface looms ever faster and the fly responds by reducing its speed to keep the rate of looming constant, resulting in a slow controlled landing [15]. The visual control of flight by sensing and responding to optic flow is an essential component of the visual ecology of

flying insects and is considered to be strongly conserved across the arthropoda [15]. Even with featureless animal coats, contrast gain would ensure that there would be some (minimal) contrast on the coat to provide the optic flow that flies rely upon for executing smooth landings [16].

More specifically, striped patterns could potentially disrupt optic flow by interfering with the radial symmetry of expanding optic flow fields via the aperture effect (see, the barber pole illusion for a human example [17–19]). This is because, in the absence of other cues, moving stripes induce the strongest motion cues in directions perpendicular to stripe orientation. To illustrate, as flies approach a zebra from a distance, they will experience radially symmetric optic flow patterns centred on the target host (figure 1bi). As the fly nears the zebra, stripes will become resolvable and occupy large parts of the visual field. These stripes will likely disrupt the pattern of optic flow so that it becomes dominated by motion cues in directions perpendicular to stripe orientation (figure 1bii). This disruption

could, in theory, be substantial enough to switch the fly from experiencing a 'landing-type' optic flow pattern to a 'translating-type' (i.e. generated when flying through the environment in the absence of a landing surface) [20], thereby fooling the fly into no longer perceiving the zebra as a surface on which to land. Alternatively, the optic flow could be disrupted by spatio-temporal aliasing. In this case, regular stripes viewed in motion could become misregistered, producing false motion directions and magnitudes [14,21] (see, wagon-wheel effect for a human example; figure 1biii), and this contradictory movement could work against the tendency for the fly to fixate the pattern, creating positive feedback and causing it to turn away. Both aperture effect and spatio-temporal aliasing within the optic flow field could, theoretically, prevent the fly from landing properly or at all on a striped surface.

On the other hand, stripes might act more centrally at the decision-making level of host-finding behaviour in horseflies. Little is known about the precise cues used by horseflies to visually segment their scene into host versus background and how this may feed into their in-flight decisions (but see [22–24]). However, it remains likely that at distances greater than 2 m from a zebra, black and white stripes fall below the resolving power of the tabanid eye (based on an estimated ommatidial acceptance angle ($\Delta\rho$) of 1° (MJ How 2019, unpublished data) and an average stripe width of 35 mm [25]; figure 1$c_i$). Tabanids seeking a bloodmeal from a zebra will initially be attracted to a grey host from a distance because the angular spatial frequency of the stripes will be higher than the cut-off angular spatial frequency of the modulation transfer function of the flies' visual system (determined by $\Delta\rho$). During the final moments of approach (between approximately 0.5–1 m from the horse [13]), the appearance of the target will change to reveal the black and white striped pelage (figure 1cii). This visual transformation could somehow disrupt the tabanid's decision-making process, perhaps by diverging away from an expected search image or by obscuring the visual coherence of the single target host [23,26].

Our prior observation that tabanids fail to slow down when approaching a zebra compared to a uniformly coloured horse [13] does not help us to discriminate between any of these hypothesized mechanisms for repelling biting flies. Here, we sought to determine whether the first of these ideas, the disruption of the radial symmetry of optic flow via the aperture effect, is responsible for the anti-biting fly mechanism of zebra stripes. To achieve this, we went into the field and employed a realistic set of targets in relation to zebra size and movement by placing uniform and patterned cloth rugs on domestic horses and investigated the behaviour of free-flying tabanid flies in their vicinity. Crucially, we included a checked rug (which would generate radially symmetric optic flow patterns) as well as horizontally and vertically striped patterns to separate the aperture effect from the other hypotheses. The aperture effect hypothesis would predict no reduction in landing rate on a checked rug compared to a grey rug of the same average luminance because the radial symmetry of optic flow would be roughly naturalistic under both circumstances. If the aperture effect were responsible for landing avoidance, then the effect would be expected only for the striped patterns, and not the checked, grey, or black patterns. The remaining hypotheses would each predict a reduction in landings for both striped and checked rugs once their patterns are resolved.

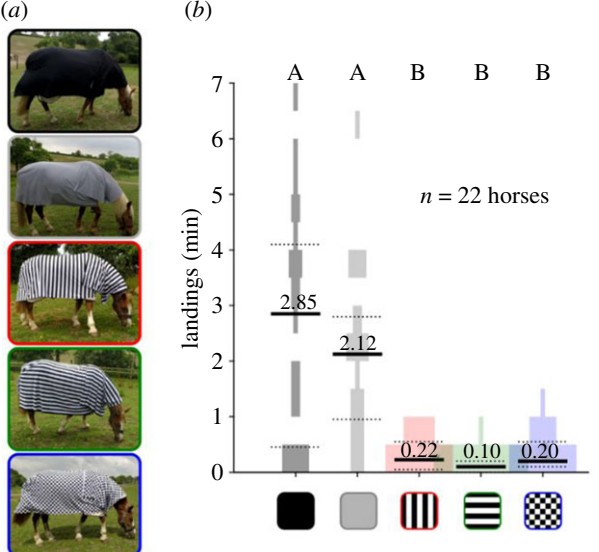

**Figure 2.** (a) Custom-made fabric pinned over commercial horse rugs. Fabric was printed with uniform black, uniform grey, vertical stripe, horizontal stripe, and checked patterns. (b) Horsefly landing rate on each of the five patterned rugs. Black lines = median, dotted lines = 25th and 75th percentile, shaded areas = violin plot of the data. Letters above each column indicate which are statistically different (Friedman's test with multi-compare and Bonferroni correction, $p < 0.05$).

## 2. Methods

Rolls of fabric (Jersey Stretch 190gsm) were custom printed (Contrado, London, UK) with five patterns: (i) uniform black, (ii) uniform grey, (iii) 3.5 cm width vertical stripes, (iv) 3.5 cm width horizontal stripes, and (v) 3.5 cm width checkerboard (figure 2a). Print shades were chosen so that all patterns, excluding black, had approximately the same average luminance (electronic supplementary material, figure S1). Patterned rugs were assembled by sewing together the printed fabric to make a $2.0 \times 2.0$ m square covering the horse's body and a smaller $0.8 \times 0.5$ m rectangle covering the neck and withers. Rugs were placed in random order on 22 different horses between 13 and 20 June 2018 at Hill Livery, Dundry, North Somerset, UK (see [13] for details) sequentially for 20 min each and affixed to underlying fly rugs with safety pins. One observer recorded the number of tabanid flies landing on one side of the rug for 20 min while another observer filmed flies approaching the other side of the horse. Usually, several different flies landed on horses per observation period, although repeat landings by single horseflies were documented from the flight trajectories [13]. To minimize the number of repeat measurements from single flies for each rug type, sampling effort from the videos was spread as evenly as possible over the full seven-day duration of the video-recording period.

Filming was conducted using a custom-built stereo camera rig consisting of two digital video cameras (Hero 5, GoPro, San Mateo, USA) affixed at either end of a 1.0 m metal bar mounted on a tripod. Cameras were positioned to approximately maintain their horizontal and vertical axes relative to the outside world and were triggered simultaneously by a single wireless remote and temporally synchronized using the audio channel. The relative position of each camera and the distortive effects of the camera optics were determined via a calibration routine involving a flat checkerboard standard and the stereo camera toolbox in Matlab (2018a, Mathworks, Natick, USA). Fly activity around horses was filmed at 60 frames per second, at a resolution of $2704 \times 1520$ pixels. Horsefly (*Haematopota pluvialis* and *Tabanus bromius*) trajectories were manually extracted from the stereo video recordings, first by identifying tabanids approaching the horse, then by manually digitizing the fly's location frame-by-frame in the pair of stereo images using a

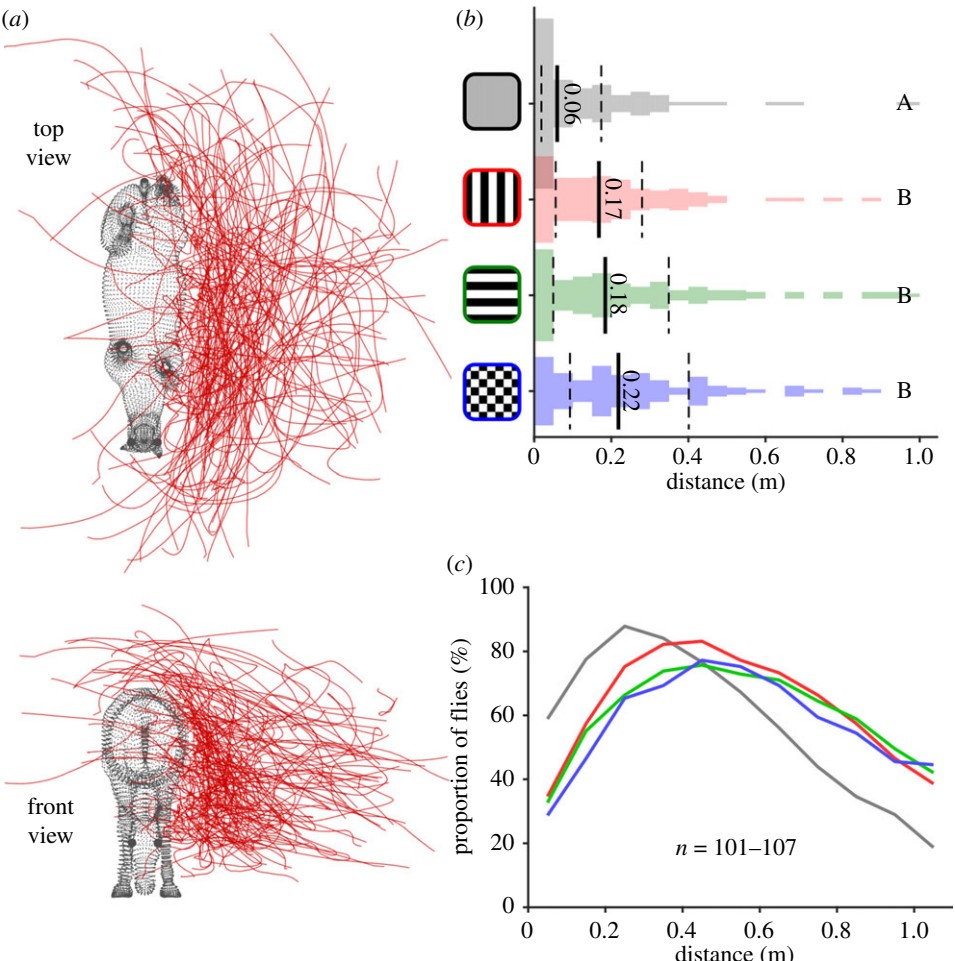

**Figure 3.** Fly distance from horses wearing patterned rugs. (*a*) Dorsoventral (top) and anteroposterior (bottom) views of 107 fly trajectories plotted around the horse location (see also electronic supplementary material, figure S2 and movie S3). Each red line shows a single horsefly trajectory around horses wearing (in this case) the vertically striped rug. (*b*) Closest recorded distance of each fly to the horse (figure 2 for conventions). (*c*) Proportion of digitized fly trajectories at different distances from the host target, colour coded according to (*b*) (*n* = 101–107 fly trajectories per rug type).

custom-written script in Matlab. Trajectories in which the horse was moving were discarded. Finally, the moment-to-moment position of the fly was extracted using the built-in functions of the stereo camera toolbox. Reconstructions of the three-dimensional flight trajectories were then smoothed with a three-point moving average. To place the trajectory data in the context of the target host, a three-dimensional mesh model of an average-sized horse was placed in the virtual space alongside the fly trajectory and manoeuvred to align with markers digitized on the front and rear regions of the original horse. Care was taken to sample the 100 or so trajectories from each rug type across as many horses as possible, over as wide time intervals as possible to further reduce the unlikely event of digitizing the same fly. Digitized segments were divided for further analysis into slow/fast flight and straight/turning flight by determining whether pairs of trajectory points fell below (slow) or above (fast) the 65th percentile speed value of 2.35 m s$^{-1}$ and whether triplets of points showed an angular turn rate of above or below the 65th percentile value of 390°/s in azimuth or 210°/s in elevation. These thresholds were chosen arbitrarily, based on the shape of histograms of the complete dataset (see results), but it must be noted that percentile thresholds between 50 and 70 all produce similar results. The overall number of trajectories analysed for each rug type was grey = 107; vertical = 101; horizontal = 107; and checked = 101.

## 3. Results

We found that the rate of horsefly landings differed dramatically according to rug pattern (figure 2*b*; Friedman's

test: *n* = 22, d.f. = 4, Chi-sq = 58.8, *p* < 0.001). Most landings were on black and grey rugs, an average of 2–3 min$^{-1}$, and these rates were significantly greater than landing rates on patterned rugs that averaged approximately 0.2 min$^{-1}$ (*post hoc* 'multi-compare' with Bonferroni correction *p* < 0.001). There were no differences in landing frequencies between either of the uniform rugs (*p* = 1), or between patterned rugs (*p* = 1).

Next, we used the reconstructed three-dimensional trajectories (figure 3*a* and electronic supplementary material, figures S2–S3) to examine horsefly behaviour around rugs with equivalent luminance (i.e. excluding the black rug). Flies approached closer to horses wearing grey rugs than to those wearing patterned rugs (figure 3*b*; Kruskal–Wallis, *p* < 0.01), but flies approached all three patterned rugs to approximately the same distance (*p* > 0.46). A similar finding was also evident in the proportion of flies approaching horses as a function of distance; fewer flies approached to within 30 cm of the patterned rugs than the grey (figure 3*c*).

When we examined flight speeds as horseflies approached rugs, we found that flies flew more slowly towards grey than towards the three patterned rugs, particularly when they were between 0.1 and 0.6 m away from the host (Kruskal–Wallis with multi-compare *post hoc* and Bonferroni correction for 0.1 to 0.6 m data, *p* < 0.05; figure 4*a*). Flight speeds were similar across the three patterned rugs, although there was some indication that flies approached horizontally striped rugs quicker than vertical (figure 4*b*; Kruskal–Wallis with

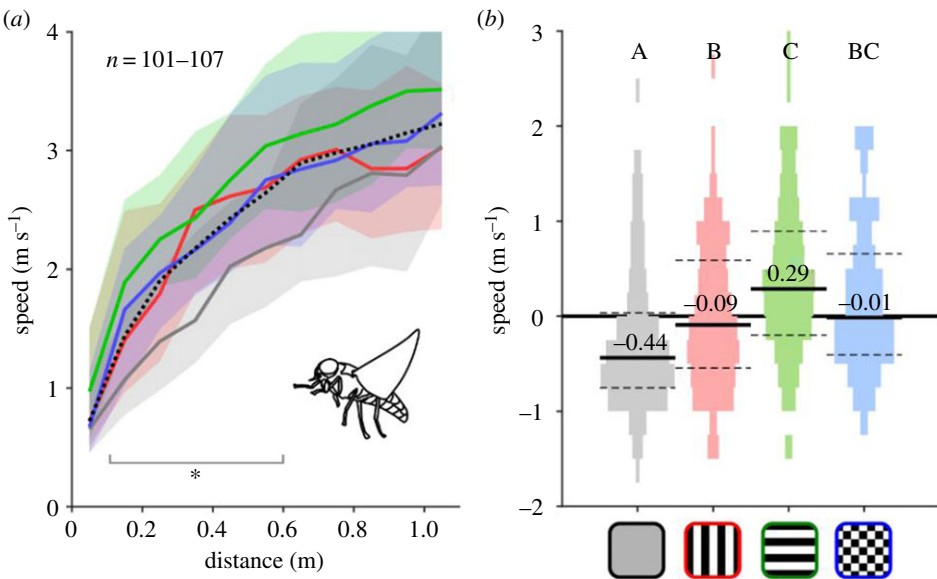

**Figure 4.** Approach speed of flies to each rug type. (*a*) Speed versus distance during fly approach, colour coded according to (*b*). Black dotted line is the median approach speed for the whole dataset. Shaded areas are the 25th and 75th percentile ranges. Star indicates the range over which grey significantly differs from the patterned rugs (Kruskal–Wallis multi-compare *post hoc* with Bonferroni correction, $p < 0.05$). See electronic supplementary material, table S1 for more statistics. (*b*) Approach speed relative to overall median. See figure 2 for graph conventions.

multi-compare *post hoc* and Bonferroni correction, vertical versus horizontal, $p = 0.011$; and electronic supplementary material table S1).

When we examined slow turns (defined as less than 65th percentile speed and greater than 35th percentile turn rate; figure 5*a*, red and pink areas) and fast turns (greater than 65th percentile speed and greater than 35th percentile turn rate; figure 5*a*, blue and pink areas) by rug type, significant differences emerged both for the number of turns per unit time and the distance at which turns were executed in relation to the horse. Slow turns were significantly more frequent near the grey than near the patterned rugs, whereas fast turns were significantly less frequent around grey than the striped rugs (figure 5*c*). This indicates that stripes (and to a lesser extent, checks) precipitated higher rates of fast manoeuvres compared to grey. Looking at the distance at which turns occurred from the animal, slow turns occurred at shorter distances from the grey rug than the checked rug, and fast turns occurred at similar distances (median roughly 0.5 m) from all four rugs (figure 5*d*). To summarize, slow turns occurred at lower rates around patterned rugs, fast turns at higher rates around striped rugs, and slow turns occurred further away from checked rugs than grey rugs.

Next, we investigated whether horsefly flight behaviour was influenced by the orientation of the stripe patterned rugs by analysing the relative levels of horizontal and vertical flight components. First, we determined the relative contribution of horizontal and vertical flight speeds by calculating a ratio between the two. As the flies approached horses, the horizontal : vertical flight speed ratio decreased, implying that they incorporated more vertical movements at closer range (figure 6*a* and electronic supplementary material, figures S2–S3). Similarly, we extracted the absolute elevation angle of horsefly flight, which again showed flight trajectories deviating from the horizontal as flies approached the horses (figure 6*b*). Nonetheless, we found no significant differences in either measure across the four rugs (Kruskal–Wallis, Chi-Sq < 2.85, $p > 0.42$) (figure 6*c*).

## 4. Discussion

We used rugs placed on domestic horses to investigate the mechanisms underlying the influence of black and white stripes on tabanid fly behaviour. Specifically, we devised tests to separate possible disruption of optic flow by the aperture effect from other theoretical mechanisms. Five independent lines of evidence indicate that the aperture effect cannot be responsible for repelling biting flies around striped patterns.

First, we saw significantly fewer landings on the checked than on the grey rug and no difference in landing frequency between checked and striped rugs (figure 2*b*). Given that the aperture effect relies upon the presence of high-contrast stripes across large parts of the visual field, checked patterns should not disrupt optic flow (and therefore visually guided flight control) through this mechanism. The observation that checked rugs were as effective as striped rugs at repelling fly landings strongly implies that the aperture effect cannot be the underlying mechanism for this effect. This result parallels that of Blaho *et al.* [27] who found that other unstriped black and white patterns (large 'blobs') also reduced tabanid landing rates on half-size herbivore models. Taken together, these findings indicate that stripes are not a necessary pattern to deter landings, but that other contrasting white and black patterns can be equally as effective.

Second, the distances to which flies approached the different rug patterns (figure 3) refute the aperture effect hypothesis, because no differences were observed between checked and striped rugs. For these patterned rugs, the proportion of flies near the horse drops off rapidly at approximately 0.45 m from the target, while flies approached much closer to grey rugs. The aperture effect hypothesis would predict that flies would approach nearer to both checked and grey rugs, with some making a controlled landing, but that flies around striped rugs would make fast uncontrolled approaches, many of which resulting in abortive turning manoeuvres, or even colliding with the host.

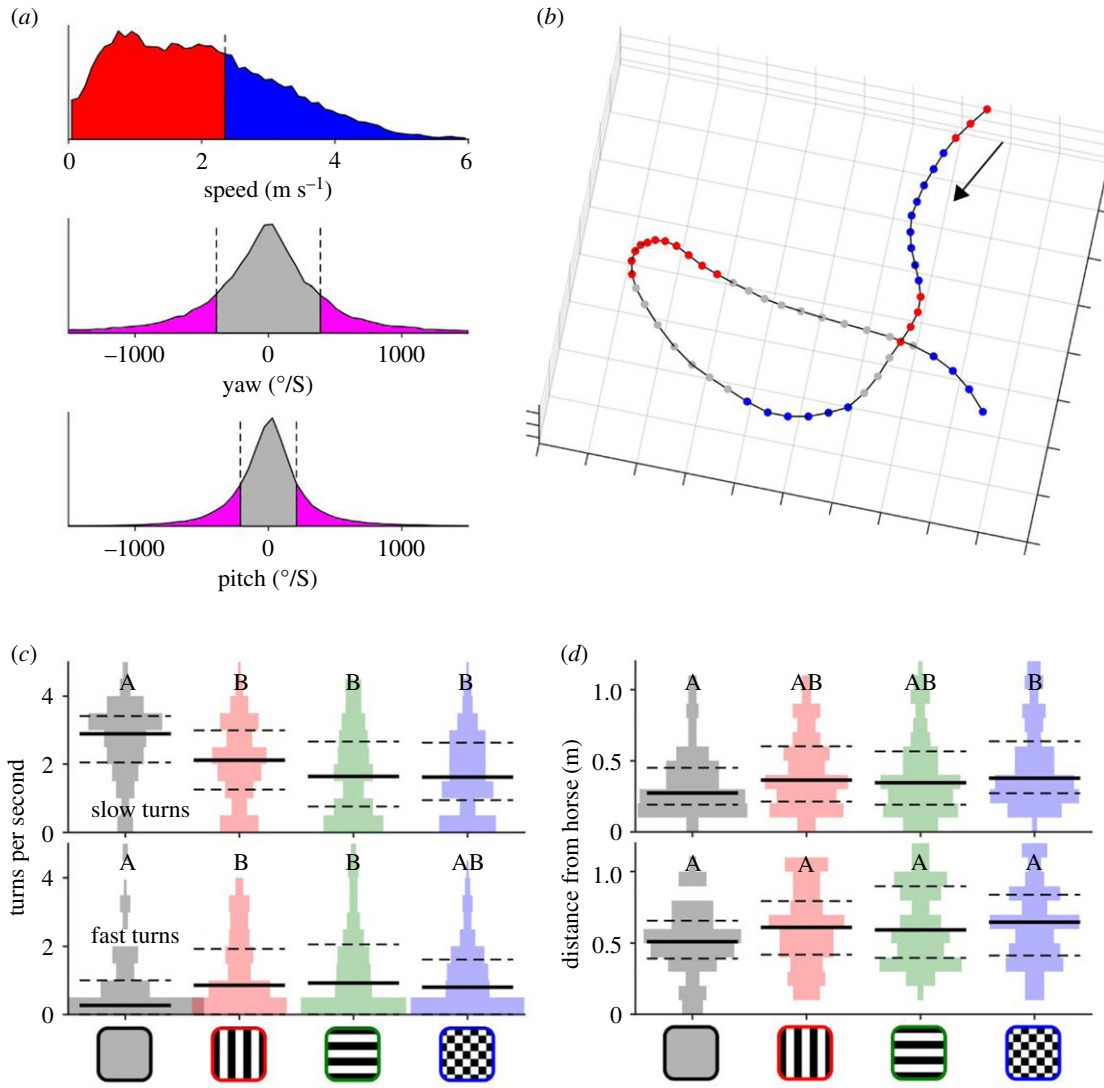

**Figure 5.** Fly turning behaviour around horses wearing patterned rugs. (*a*) Histograms of speed, pitch, and yaw of pooled data demonstrating the cut-off points used to define slow/fast flight (red and blue, respectively) and turning/straight flight (pink and grey, respectively). (*b*) Example flight segmented into fast (blue) and slow (red) turns viewed from above. Black arrow indicates direction of flight. (*c*) Rate of slow (top) and fast (bottom) turns around the four rugs. (*d*) Distance of slow and fast turns from horses. See figure 2 for graph conventions for *c* and *d*.

Third, our data showed that flies approached striped and checked rugs equally fast (figure 4). If the aperture effect were responsible for interfering with fly landing (for example by inducing enough radial asymmetry in the optic flow pattern to 'switch' the fly from sensing a landing-type optic flow pattern to a translational-type pattern [20]), then we would expect flies to fly faster around striped rugs than around the checked rug. Of course, it is also possible that flies performed fast flights around the patterned rugs simply because the high-contrast features generated stronger and more reliable motion cues, allowing the flies to execute better-controlled manoeuvres more quickly. This seems unlikely though, given that horseflies are almost certainly able to modulate their visual contrast gain to detect reliable optic flow around the grey rugs [16], and also because very few flights near the checked rugs actually ended in landings.

Fourth, our findings regarding turn speeds (figure 5) again refute the aperture effect hypotheses, because all three patterned rugs caused a reduction in the number of slow measured turns compared to the grey rug. The aperture effect hypothesis would predict no difference between checked and grey rugs, but this was not observed. Regarding fast turns, flies appeared to rapidly turn away from stripes

once they were resolved rather than continuing to fly quickly as predicted by the aperture effect hypothesis. Furthermore, slow turns occurred closer to the grey rug than to the checked rug, running counter to the aperture effect hypothesis, which would predict that any turns would occur at similar distances for grey and checked rugs, assuming that the grey rug displays at least a low contrast and that movement sensing neurons in the fly's visual system signal optic flow correctly, independently of the contrast or texture of the image.

Fifth, the aperture effect mechanism would predict differences in flight paths with respect to stripe orientation. Flies might be expected to fly parallel to stripe orientation due to the lower levels of visual contrast detected in these directions (caused by the lower number of edges in parallel versus perpendicular to the stripes). Yet we found no difference in the ratio of horizontal to vertical flight movements between horizontal and vertically striped rugs (figure 6).

These multiple lines of evidence force us to conclude that stripes, characteristic of zebras, do not reduce tabanid landings by disrupting optic flow via the aperture effect. However, we cannot yet distinguish alternative possibilities that may affect flight control or decision-making in horsefly host-finding behaviour. Regarding flight control, a remaining possibility is

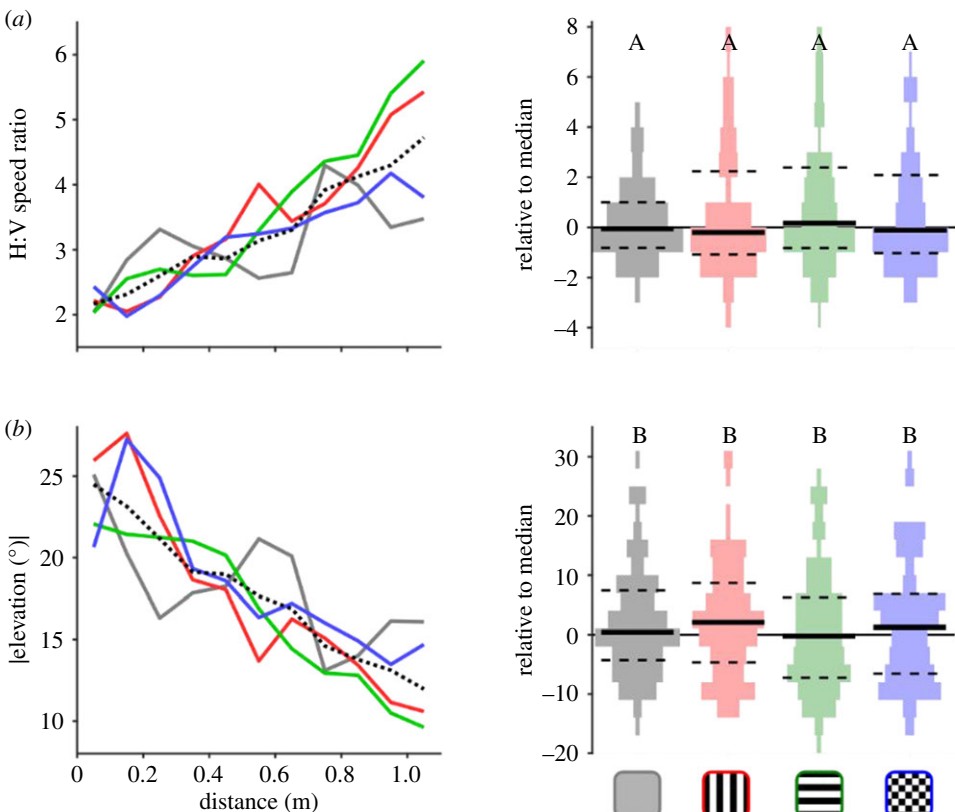

**Figure 6.** Horizontal and vertical flight characteristics around patterned rugs. Ratio between horizontal and vertical flight speeds (*a,b*) and absolute elevation angle (*c*) relative to distance and (*d*) overall median, figure 2 for graph conventions for graphs right.

that stripes and other regular patterns such as checked rugs may induce visual aliasing that disrupt the optic flow. The spatial period of the stripe and checkerboard patterns used in our experiment was 7.0 cm, which, at a viewing distance of 1 m, would correspond to an angular period of about 4°. Thus, spatio-temporal aliasing could occur at this distance if the inter-ommatidial angle falls near to half this value. This value of the interommatidial angle is approximately realistic for horseflies (MJ How 2019, unpublished data) and could explain why fast turns were initiated at distances of approximately 70 cm for the patterned rugs (figure 5*d*). A necessary prerequisite for aliasing is that the pattern must have regular repeating elements that may be misregistered when seen in motion. Future experiments will need to compare the effects of rugs with regularly and irregularly spaced high-contrast elements.

Alternatively, if high-contrast patterns affect the higher level decision-making of flies when finding hosts, then we would expect many different types of high-contrast patterns (varying in distribution and spatial frequency) to be equally effective at repelling the attention of biting flies. It is important to note that stripes, checks, and large irregular contrasting circles [27] all hinder tabanids from landing on host targets; there is nothing special about stripes. This raises the question of why zebras have horizontal and vertical stripes *per se* rather than contrasting black and white pelage of another pattern.

Although the mechanism by which stripes are formed *in utero* is partially understood [28–30], we suggest that developmental biologists turn their attention to ease of pattern formation in these and other mammalian species [31].

Before concluding, we want to emphasize that stripes are only one way by which zebras reduce successful probing for blood by biting flies. Others include behavioural means including frequent swishing of tails and running away from tabanid annoyance [13] and zebra skin odour deterring tsetse flies from landing [32], suggesting that there are severe selection pressures for African Equidae to avoid biting fly attack [25].

Data accessibility. All data not presented in the manuscript, including raw stereo video files, digitized fly location data, and Matlab code are available from the Dryad Digital Repository: https://doi.org/10.5061/dryad.18931zctk [33].

Authors' contributions. T.C. and M.J.H. conceived and designed the study and prepared the manuscript. All authors collected field data and performed preliminary analyses. M.J.H. analysed the fly trajectory data.

Competing interests. We declare we have no competing interests.

Funding. We thank the Institute for Advanced Studies, University of Bristol, and the Royal Society (grant no. UF140558) for financial support.

Acknowledgements. We thank Terri Hill and the Hill Livery team for logistical support and Nick Scott-Samuel for help in the field. We thank the two referees for helpful comments on the manuscript.

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
