## [Reviewer comments · Proceedings of the Royal Society B: Biological Sciences]

Review History

RSPB-2020-0611.R0 (Original submission)

Review form: Reviewer 1

Recommendation

Major revision is needed (please make suggestions in comments)

Scientific importance: Is the manuscript an original and important contribution to its field?

Marginal

General interest: Is the paper of sufficient general interest?

Excellent

Quality of the paper: Is the overall quality of the paper suitable?

Acceptable

Is the length of the paper justified?

Yes

Should the paper be seen by a specialist statistical reviewer?

No

Do you have any concerns about statistical analyses in this paper? If so, please specify them explicitly in your report.

No

It is a condition of publication that authors make their supporting data, code and materials available - either as supplementary material or hosted in an external repository. Please rate, if applicable, the supporting data on the following criteria.

Is it accessible?

N/A

Is it clear?

N/A

Is it adequate?

N/A

Do you have any ethical concerns with this paper?

No

Comments to the Author

Review: How do zebra stripes prevent landings by tabanid biting flies?

In this paper, the authors aim to investigate the function of zebra stripes and their apparent repellent effect on biting flies by exploring the mechanism by which high contrast stripes prevent landings by flies. They do this by covering horses with a range of high contrast patterns and recording landings and flight trajectories of approaching flies. The subject is of great general interest, the study is well-written, and the methodology is thorough. I enjoyed reading this paper but there appear to be two main conceptual weaknesses that severely limit the validity of the conclusions drawn and these are related to the hypothesis that, as the flies approach the high contrast patterns, they suddenly are 'revealed' to the flies and that this disrupts their targeting system when landing.

I do not understand the 'reveal' hypothesis, which the authors conclude, is the most likely mechanism preventing biting flies from landing on the tested high contrast patterns. This is because, I disagree that there is good optical evidence that, as the flies approach a pattern that becomes resolveable, it will be 'suddenly' visible in high contrast as the authors suggest on line 63 and throughout the text. I would expect that, rather than appearing suddenly, the stripes gradually become resolveable and increase in contrast as the flies approach, first being perceived as low contrast and then becoming higher contrast and lower spatial frequency. This is mainly due to optical principals such as the contrast transfer function: that is, that the contrast of the gratings that can be resolved by an optical system decrease as they get finer. I might have totally misunderstood what the authors meant with this reveal hypothesis, in which case the underlying principals should be more clearly explained in the text. But I nonetheless find it difficult to understand how a pattern that becomes resolveable on approach will 'suddenly' appear.

I have difficulty understanding the logic behind the interpretation of how the different rug types affect the flies' flight manoeuvres. I do not follow the logic of the explanation given on line 220. Why would the high contrast patterns cause the flies to turn away? This point is not answered clearly. The flies could make faster turns around the patterned rugs than around the grey rug because the high contrast edges provide them with more reliable visual cues that they can use to better control fast flight. The authors should be careful when interpreting the effect of the grey rug, as it could potentially provide the flies with just as salient optic flow cues due to contrast gain enhancements (Harris et al. 2000).

Harris, R. A., O'Carroll, D. C. & Laughlin, S. B. (2000) Contrast gain reduction in fly motion

adaptation. Neuron 28:595-606.

I do not understand the reasoning given on line 225. As the flies approach a horizontal stripe, they will still experience expansion cues at the edges. If they balanced these expansion cues, they could still make effective 'optic flow'-based controlled landings.

To summarise, the logic behind the two main proposed hypotheses is not well supported in the text. As such, I do not find that the interpretations are supported by the results. To address this issue, the authors need to explain the mechanisms that they are testing more clearly and from the basic principles of vision.

Review form: Reviewer 2

Recommendation

Major revision is needed (please make suggestions in comments)

Scientific importance: Is the manuscript an original and important contribution to its field?

Marginal

General interest: Is the paper of sufficient general interest?

Excellent

Quality of the paper: Is the overall quality of the paper suitable?

Acceptable

Is the length of the paper justified?

Yes

Should the paper be seen by a specialist statistical reviewer?

No

Do you have any concerns about statistical analyses in this paper? If so, please specify them explicitly in your report.

No

It is a condition of publication that authors make their supporting data, code and materials available - either as supplementary material or hosted in an external repository. Please rate, if applicable, the supporting data on the following criteria.

Is it accessible?

N/A

Is it clear?

N/A

Is it adequate?

N/A

Do you have any ethical concerns with this paper?

No

Comments to the Author

GENERAL

The functional role of the black-and-white stripes on zebra coats has been a longstanding enigma. This interesting study builds on a number of earlier studies that have examined the visual effect(s) that the stripes may have in deterring parasitic flies, such as tsetse flies and tabanids, from landing on the coat. A number of similar earlier studies (cited in the present manuscript, some of which come from the same laboratory) have already provided clear evidence that these flies avoid striped coats, and prefer to land on coats that do not carry stripes. The present study confirms these earlier findings - based on detailed analysis of flight speeds, approach distances and turning rates - and takes one further (small) step by examining the reaction to checkerboard patterns. It is found that checkerboard patterns have the same repelling effect as stripes.

The authors interpret this finding to mean that the deterrent effect of stripes is not due to "disruption" of optic flow (caused by an aperture effect), but to what they describe as a phenomenon of "revelation". It is true that stripes generate an unusual pattern of optic flow that is different from what would normally be encountered by a landing insect. However, I do not see a strong a priori reason to suspect that this may be the cause - given that many insects, including bees, show no tendency to avoid landing on striped patterns.

While the disruption hypothesis is still plausible, I am not sure that the experimental findings exclude other possibilities. For example, any periodic pattern - be it a linear grating or a checkerboard - would be subject to spatial aliasing when it is viewed by the insect's compound eye. When the (angular) period of the stripes (λ) is larger than twice the interommatidial angle ($\Delta\phi$), the stripes (or checkerboard) would appear to move in the same direction in the insect's eye as they do in the real world. However, when λ is smaller than twice $\Delta\phi$ (which will occur when the insect is far away), the stripes (or checkerboard) would appear to move in the opposite direction. This effect of spatial aliasing in the fly's eye was demonstrated by Goetz and Reichardt back in the sixties. When the fly is very far away from the pattern, the pattern will appear grey because its (angular) spatial frequency is higher than the cutoff angular spatial frequency of the modulation transfer function (MTF) of the visual system, which is determined by the width ($\Delta\rho$) of the angular acceptance function. (The optics functions as a spatial low-pass filter). As the fly approaches the pattern the angular period of the pattern will progressively increase, and the apparent contrast of the pattern will increase to the point where the pattern becomes visible (i.e. the contrast is suprathreshold). If the angular period of the pattern at this distance is smaller than twice the interommatidial angle, then spatial aliasing would occur and the apparent movement of the pattern would be in a direction opposite to that which would occur in a normal visual environment - which is dominated by low, non-aliasing spatial frequencies. This contradictory movement would work against the tendency for the fly to fixate the pattern, creating positive feedback and causing it to turn away from the pattern. For the stripe and checkerboard patterns used in these experiments, the spatial period is 7.0 cm, which, at a viewing distance of 70 cm, would correspond to an angular period of 0.1 radians, or about 5.7 degrees. Thus, spatial aliasing would occur at this distance if the interommatidial angle is larger than half this value (2.85 degrees). This value of interommatidial angle is not unrealistic for flies, and the distance of 70 cm for fast turns corresponds approximately to the values shown in Fig 5d. That is not to say that spatial aliasing is the correct explanation for the findings - it is just an additional possibility that the authors may wish to consider.

I find it difficult to understand the 'reveal' hypothesis that is offered as an explanation for the present findings. I can see how the stripes and the checkerboard patterns - which cannot be resolved at large distances and will therefore appear grey - will suddenly be 'revealed' to the insect when it comes closer. But what about coats that carry random spots, blobs or patches, which also seem to repel the flies (see, for example, Blaho et al. 2012)? Such patterns would be at least partially visible at all distances - there would be no sudden 'revelation', as far as I can see. With the textureless coats, there would have to be some (minimal) contrast on the coat, to provide the optic flow that flies rely upon for executing smooth landings. Now, if we are saying that there

is no sudden ‘revelation’ in these ‘textureless’ coats, does this imply that the low-contrast textures in such coats have the same appearance and spatial properties at all scales of magnification – rather like a fractal pattern, or a scene with a power spectrum of $[1/(\text{spatial frequency})^2]$? If that is what the authors are saying, then, according to the ‘reveal’ hypothesis, the flies should show no hesitation towards landing on a high-contrast fractal pattern. This prediction could be tested quite easily.

As far as I can see, this study does indeed thoroughly exclude one possibility – namely, that the avoidance of stripes on a zebra is not due to the disruption of the expected optic flow pattern, caused by the aperture effect. But in doing so, the results and their interpretation raise more questions, and do not seem to provide an answer for the reason why these flies avoid landing on an animal’s coat whenever it carries *any* pattern – be it stripes, checkerboards, or spots or patches. The ‘appearance’ explanation therefore does not make sense to me, because it does not seem to provide a logically consistent explanation for all of the observed behaviors that have been documented in the literature. It seems to me that this calls for more experimental work using a variety of different patterns.

SPECIFIC

Lines 80- 83: This statement would not be true if the grey rug has zero contrast, because it would not generate any optic flow. Need to specify the properties that you assume a ‘grey’ rug to have. Same issue with the statement in lines 86-88.

Lines 99-100: I presume the flies are not marked, so they cannot be identified individually. Can you provide some idea of how many flies were likely to be participating in the experiments, so that one can get a feel for the statistical robustness of the data? Landings from multiple individuals would be preferable to multiple landings from a single fly, because the former would enhance statistical independence.

Legend to Figure 3: When you say $n=107$ flies or $n=101-107$ flies per rug type, I presume you the number of flights analyzed, and not the number of individual flies.

Figure 3a: Might be useful to show trajectories for other patterns too, including the grey rug, in the Supplementary Information section.

Figure 3b: Not clear what you mean by ‘stacked histogram’ here. This term usually applies to histograms in which bars are stacked vertically on top of each other, with each bar representing the count of a different variable within the same bin. That does not seem to be the case here, and it is unclear what the bars mean. Is it simply a histogram showing bin counts at various distances? Same thing in Fig. 4b: Bin counts at various speeds? Such plots are usually termed ‘violin plots’.

Fig. 5b and legend: Is the flight segmented into fast and slow speeds, or fast and slow turning rates? Please clarify. The colors seem to correspond to those used in the speed histogram shown in Fig. 5a, and not to those used in the yaw and pitch histograms.

Lines 223-224, where you say “an optic flow hypothesis would predict any turns would occur at similar distance for grey and checked rugs”: This would be true only if (a) the grey rugs display at least a low contrast and (b) the movement sensing neurons in the fly’s visual system signal optic flow (image speed) veridically, independently of the contrast or texture of the image.

Decision letter (RSPB-2020-0611.R0)

12-May-2020

Dear Dr How:

I am writing to inform you that your manuscript RSPB-2020-0611 entitled "How do zebra stripes prevent landings by tabanid biting flies?" has, in its current form, been rejected for publication in Proceedings B.

This action has been taken on the advice of referees, who have recommended that substantial revisions are necessary. With this in mind we would be happy to consider a resubmission, provided the comments of the referees are fully addressed. However please note that this is not a provisional acceptance.

Sincerely,

Dr Robert Barton

Associate Editor

Comments to Author:

Two expert referees have now carefully reviewed your manuscript, and both have surprisingly similar comments and suggestions. Both agree that your study is well written, topical and very interesting but unfortunately both feel that your main hypotheses are not supported by the data, and indeed that your data possibly open more questions than they answer. Essentially, both reviewers are of the opinion that due to the fact that you have not accounted for the optics of the compound eye - and its implications for aliasing and contrast in the optic flow experienced - disruption of the expected optic flow might not be the only explanation for your fly avoidance data (although it is definitely one possible explanation). Additionally both referees have difficulty with your "reveal" hypothesis, which the second referee suggests would require additional experiments, with a variety of other patterns (such as fractal patterns), to demonstrate. Unfortunately, you will likely need to spend a considerable amount of effort to address the concerns of the referees before the manuscript is ready for publication.

Reviewer(s)' Comments to Author:

Referee: 1

Comments to the Author(s)

Review: How do zebra stripes prevent landings by tabanid biting flies?

In this paper, the authors aim to investigate the function of zebra stripes and their apparent repellent effect on biting flies by exploring the mechanism by which high contrast stripes prevent landings by flies. They do this by covering horses with a range of high contrast patterns and recording landings and flight trajectories of approaching flies. The subject is of great general interest, the study is well-written, and the methodology is thorough. I enjoyed reading this paper but there appear to be two main conceptual weaknesses that severely limit the validity of the conclusions drawn and these are related to the hypothesis that, as the flies approach the high contrast patterns, they suddenly are 'revealed' to the flies and that this disrupts their targeting system when landing.

I do not understand the 'reveal' hypothesis, which the authors conclude, is the most likely mechanism preventing biting flies from landing on the tested high contrast patterns. This is because, I disagree that there is good optical evidence that, as the flies approach a pattern that becomes resolveable, it will be 'suddenly' visible in high contrast as the authors suggest on line 63 and throughout the text. I would expect that, rather than appearing suddenly, the stripes gradually become resolveable and increase in contrast as the flies approach, first being perceived as low contrast and then becoming higher contrast and lower spatial frequency. This is mainly due to optical principals such as the contrast transfer function: that is, that the contrast of the gratings that can be resolved by an optical system decrease as they get finer. I might have totally misunderstood what the authors meant with this reveal hypothesis, in which case the underlying principals should be more clearly explained in the text. But I nonetheless find it difficult to understand how a pattern that becomes resolveable on approach will 'suddenly' appear.

I have difficulty understanding the logic behind the interpretation of how the different rug types affect the flies' flight manoeuvres. I do not follow the logic of the explanation given on line 220. Why would the high contrast patterns cause the flies to turn away? This point is not answered clearly. The flies could make faster turns around the patterned rugs than around the grey rug because the high contrast edges provide them with more reliable visual cues that they can use to better control fast flight. The authors should be careful when interpreting the effect of the grey rug, as it could potentially provide the flies with just as salient optic flow cues due to contrast gain enhancements (Harris et al. 2000).

Harris, R. A., O'Carroll, D. C. & Laughlin, S. B. (2000) Contrast gain reduction in fly motion adaptation. *Neuron* 28:595-606.

I do not understand the reasoning given on line 225. As the flies approach a horizontal stripe, they will still experience expansion cues at the edges. If they balanced these expansion cues, they could still make effective 'optic flow'-based controlled landings.

To summarise, the logic behind the two main proposed hypotheses is not well supported in the text. As such, I do not find that the interpretations are supported by the results. To address this issue, the authors need to explain the mechanisms that they are testing more clearly and from the basic principles of vision.

Referee: 2

Comments to the Author(s)

GENERAL

The functional role of the black-and-white stripes on zebra coats has been a longstanding enigma. This interesting study builds on a number of earlier studies that have examined the visual effect(s) that the stripes may have in deterring parasitic flies, such as tsetse flies and tabanids, from landing on the coat. A number of similar earlier studies (cited in the present manuscript, some of which come from the same laboratory) have already provided clear evidence that these flies avoid striped coats, and prefer to land on coats that do not carry stripes. The present study confirms these earlier findings - based on detailed analysis of flight speeds, approach distances and turning rates - and takes one further (small) step by examining the reaction to checkerboard patterns. It is found that checkerboard patterns have the same repelling effect as stripes.

The authors interpret this finding to mean that the deterrent effect of stripes is not due to "disruption" of optic flow (caused by an aperture effect), but to what they describe as a phenomenon of "revelation". It is true that stripes generate an unusual pattern of optic flow that is different from what would normally be encountered by a landing insect. However, I do not see a strong a priori reason to suspect that this may be the cause - given that many insects, including bees, show no tendency to avoid landing on striped patterns.

While the disruption hypothesis is still plausible, I am not sure that the experimental findings exclude other possibilities. For example, any periodic pattern - be it a linear grating or a checkerboard - would be subject to spatial aliasing when it is viewed by the insect's compound eye. When the (angular) period of the stripes (λ) is larger than twice the interommatidial angle ($\Delta\phi$), the stripes (or checkerboard) would appear to move in the same direction in the insect's eye as they do in the real world. However, when λ is smaller than twice $\Delta\phi$ (which will occur when the insect is far away), the stripes (or checkerboard) would appear to move in the opposite direction. This effect of spatial aliasing in the fly's eye was demonstrated by Goetz and Reichardt back in the sixties. When the fly is very far away from the pattern, the pattern will appear grey because its (angular) spatial frequency is higher than the cutoff angular spatial frequency of the modulation transfer function (MTF) of the visual system, which is determined by the width ($\Delta\rho$) of the angular acceptance function. (The optics functions as a spatial low-pass filter). As the fly approaches the pattern the angular period of the pattern will progressively increase, and the apparent contrast of the pattern will increase to the point where the pattern becomes visible (i.e. the contrast is suprathreshold). If the angular period of the pattern at this distance is smaller than twice the interommatidial angle, then spatial aliasing would occur and the apparent movement of the pattern would be in a direction opposite to that which would occur in a normal visual environment - which is dominated by low, non-aliasing spatial frequencies. This contradictory movement would work against the tendency for the fly to fixate the pattern, creating positive feedback and causing it to turn away from the pattern. For the stripe and checkerboard patterns used in these experiments, the spatial period is 7.0 cm, which, at a viewing distance of 70 cm, would correspond to an angular period of 0.1 radians, or about 5.7 degrees. Thus, spatial aliasing would occur at this distance if the interommatidial angle is larger than half this value (2.85 degrees). This value of interommatidial angle is not unrealistic for flies, and the distance of 70 cm for fast turns corresponds approximately to the values shown in Fig 5d. That is not to say that spatial aliasing is the correct explanation for the findings - it is just an additional possibility that the authors may wish to consider.

I find it difficult to understand the 'reveal' hypothesis that is offered as an explanation for the present findings. I can see how the stripes and the checkerboard patterns - which cannot be resolved at large distances and will therefore appear grey - will suddenly be 'revealed' to the insect when it comes closer. But what about coats that carry random spots, blobs or patches, which also seem to repel the flies (see, for example, Blaho et al. 2012)? Such patterns would be at least partially visible at all distances - there would be no sudden 'revelation', as far as I can see. With the textureless coats, there would have to be some (minimal) contrast on the coat, to provide the optic flow that flies rely upon for executing smooth landings. Now, if we are saying that there is no sudden 'revelation' in these 'textureless' coats, does this imply that the low-contrast textures in such coats have the same appearance and spatial properties at all scales of magnification -

rather like a fractal pattern, or a scene with a power spectrum of $[1/(\text{spatial frequency})^2]$? If that is what the authors are saying, then, according to the 'reveal' hypothesis, the flies should show no hesitation towards landing on a high-contrast fractal pattern. This prediction could be tested quite easily.

As far as I can see, this study does indeed thoroughly exclude one possibility – namely, that the avoidance of stripes on a zebra is not due to the disruption of the expected optic flow pattern, caused by the aperture effect. But in doing so, the results and their interpretation raise more questions, and do not seem to provide an answer for the reason why these flies avoid landing on an animal's coat whenever it carries *any* pattern – be it stripes, checkerboards, or spots or patches. The 'appearance' explanation therefore does not make sense to me, because it does not seem to provide a logically consistent explanation for all of the observed behaviors that have been documented in the literature. It seems to me that this calls for more experimental work using a variety of different patterns.

SPECIFIC

Lines 80- 83: This statement would not be true if the grey rug has zero contrast, because it would not generate any optic flow. Need to specify the properties that you assume a 'grey' rug to have. Same issue with the statement in lines 86-88.

Lines 99-100: I presume the flies are not marked, so they cannot be identified individually. Can you provide some idea of how many flies were likely to be participating in the experiments, so that one can get a feel for the statistical robustness of the data? Landings from multiple individuals would be preferable to multiple landings from a single fly, because the former would enhance statistical independence.

Legend to Figure 3: When you say $n=107$ flies or $n=101-107$ flies per rug type, I presume you the number of flights analyzed, and not the number of individual flies.

Figure 3a: Might be useful to show trajectories for other patterns too, including the grey rug, in the Supplementary Information section.

Figure 3b: Not clear what you mean by 'stacked histogram' here. This term usually applies to histograms in which bars are stacked vertically on top of each other, with each bar representing the count of a different variable within the same bin. That does not seem to be the case here, and it is unclear what the bars mean. Is it simply a histogram showing bin counts at various distances? Same thing in Fig. 4b: Bin counts at various speeds? Such plots are usually termed 'violin plots'.

Fig. 5b and legend: Is the flight segmented into fast and slow speeds, or fast and slow turning rates? Please clarify. The colors seem to correspond to those used in the speed histogram shown in Fig. 5a, and not to those used in the yaw and pitch histograms.

Lines 223-224, where you say "an optic flow hypothesis would predict any turns would occur at similar distance for grey and checked rugs": This would be true only if (a) the grey rugs display at least a low contrast and (b) the movement sensing neurons in the fly's visual system signal optic flow (image speed) veridically, independently of the contrast or texture of the image.

Author's Response to Decision Letter for (RSPB-2020-0611.R0)

See Appendix A.

RSPB-2020-1521.R0

Review form: Reviewer 1 (Emily Baird)

Recommendation

Reject – article is not of sufficient interest (we will consider a transfer to another journal)

Scientific importance: Is the manuscript an original and important contribution to its field?

Marginal

General interest: Is the paper of sufficient general interest?

Acceptable

Quality of the paper: Is the overall quality of the paper suitable?

Good

Is the length of the paper justified?

Yes

Should the paper be seen by a specialist statistical reviewer?

No

Do you have any concerns about statistical analyses in this paper? If so, please specify them explicitly in your report.

No

It is a condition of publication that authors make their supporting data, code and materials available - either as supplementary material or hosted in an external repository. Please rate, if applicable, the supporting data on the following criteria.

Is it accessible?

No

Is it clear?

N/A

Is it adequate?

N/A

Do you have any ethical concerns with this paper?

No

Comments to the Author

I appreciate the changes that the authors have made to address the concerns of the reviewers. In general, I am satisfied with the response. I just have one remaining comment and a few suggestions for changes.

Page 10 line 196. The criteria used to determine slow/fast turns should be defined in the method section. More generally, what is the biological relevance of this categorization? Without further justification, calculating these parameters seems a bit artificial. How do the specific values used for categorization (fig 5a) affect the results (figure 5c-d)? Finally, I am surprised that some of the first segments of the example trajectory given in figure 5b could be considered as turns as the start of the trajectory seems relatively straight.

object is approached is not always true (for example if the animal continuously decreases speed). I suggest specifying that the fly approaches 'at a constant speed'.

Page 3 line 91. Away from which point?

Page 5 line 143-148. Was any noise-removal or smoothing procedure involved when reconstructing the fly trajectories? If yes, this should be specified in the methods.

Page 7 line 148-149. Did the authors consider movements of the horses during or between recordings?

Page 8 figure 2b. The violin plots look a bit confusing because all the segments are very scattered. I suggest to plot instead a smoothed curve (e.g. as produced by the default `geom_violin` function in R.)

Page 10 figure 4a. Why are there two 'n'?

Page 10 line 217. How did the authors determine the vertical/horizontal axes in their coordinate system? Did they use the horse to approximate the vertical direction?

Page 10 line 221. In a similar way, what was the reference used to calculate the absolute elevation? Was it a point on the horse or on the ground?

Page 12 line 270 'run' instead of 'running' might be better here

Page 13 line 304. ...[32], suggesting 'that' there are severe...

Review form: Reviewer 2

Recommendation

Accept with minor revision (please list in comments)

Scientific importance: Is the manuscript an original and important contribution to its field?

Good

General interest: Is the paper of sufficient general interest?

Good

Quality of the paper: Is the overall quality of the paper suitable?

Good

Is the length of the paper justified?

Yes

Should the paper be seen by a specialist statistical reviewer?

No

Do you have any concerns about statistical analyses in this paper? If so, please specify them explicitly in your report.

No

It is a condition of publication that authors make their supporting data, code and materials available - either as supplementary material or hosted in an external repository. Please rate, if applicable, the supporting data on the following criteria.

Is it accessible?

N/A

Is it clear?

N/A

Is it adequate?

N/A

Do you have any ethical concerns with this paper?

No

Comments to the Author

Thank you for these revisions.

I am glad to note that the manuscript has now been revised to remove the previous conclusion about the role of the 'appearance' hypothesis, and to now conclude that the avoidance of stripes is not caused by the aperture effect. The real reason for the increased avoidance of coats decorated with stripes and checkerboard patterns (as opposed to grey and black coats) remains to be investigated, but at least the revised conclusion of the study is now compatible with the data.

In lines 117-119: I think it would be crucial to add here that, if the aperture effect were responsible for landing avoidance, then you would expect avoidance only for the striped patterns, and not for the checkerboard, grey or black patterns.

Decision letter (RSPB-2020-1521.R0)

22-Jul-2020

Dear Dr How

I am pleased to inform you that your manuscript RSPB-2020-1521 entitled "Zebra stripes, tabanid biting flies, and the aperture effect" has been accepted for publication in Proceedings B.

The referee(s) have recommended publication, but also suggest some minor revisions to your manuscript. Therefore, I invite you to respond to the referee(s)' comments and revise your manuscript. Because the schedule for publication is very tight, it is a condition of publication that you submit the revised version of your manuscript within 7 days. If you do not think you will be able to meet this date please let us know.

- 1) A text file of the manuscript (doc, txt, rtf or tex), including the references, tables (including captions) and figure captions. Please remove any tracked changes from the text before submission. PDF files are not an accepted format for the "Main Document".

2) A separate electronic file of each figure (tiff, EPS or print-quality PDF preferred). The format should be produced directly from original creation package, or original software format. PowerPoint files are not accepted.

3) Electronic supplementary material: this should be contained in a separate file and where possible, all ESM should be combined into a single file. All supplementary materials accompanying an accepted article will be treated as in their final form. They will be published alongside the paper on the journal website and posted on the online figshare repository. Files on figshare will be made available approximately one week before the accompanying article so that the supplementary material can be attributed a unique DOI.

Sincerely,

Dr Robert Barton

Associate Editor

Board Member

Comments to Author:

The two reviewers have now seen your revisions and are happy with your responses to their comments. Both feel that your study is well executed and that the conclusions now match the data that was obtained, but also acknowledge that the real reasons for the increased avoidance of coats decorated with stripes and checkerboard patterns still remains an open question. Both reviewers had a number of final minor concerns that should be addressed prior to publication.

Reviewer(s)' Comments to Author:

Referee: 2

Comments to the Author(s).

Thank you for these revisions.

I am glad to note that the manuscript has now been revised to remove the previous conclusion about the role of the 'appearance' hypothesis, and to now conclude that the avoidance of stripes is not caused by the aperture effect. The real reason for the increased avoidance of coats decorated with stripes and checkerboard patterns (as opposed to grey and black coats) remains to be investigated, but at least the revised conclusion of the study is now compatible with the data.

In lines 117-119: I think it would be crucial to add here that, if the aperture effect were responsible for landing avoidance, then you would expect avoidance only for the striped patterns, and not for the checkerboard, grey or black patterns.

Referee: 1

Comments to the Author(s).

I appreciate the changes that the authors have made to address the concerns of the reviewers. In general, I am satisfied with the response. I just have one remaining comment and a few suggestions for changes.

Page 10 line 196. The criteria used to determine slow/fast turns should be defined in the method section. More generally, what is the biological relevance of this categorization? Without further justification, calculating these parameters seems a bit artificial. How do the specific values used for categorization (fig 5a) affect the results (figure 5c-d)? Finally, I am surprised that some of the first segments of the example trajectory given in figure 5b could be considered as turns as the start of the trajectory seems relatively straight.

object is approached is not always true (for example if the animal continuously decreases speed). I suggest specifying that the fly approaches 'at a constant speed'.

Page 3 line 91. Away from which point?

Page 5 line 143-148. Was any noise-removal or smoothing procedure involved when reconstructing the fly trajectories? If yes, this should be specified in the methods.

Page 7 line 148-149. Did the authors consider movements of the horses during or between recordings?

Page 8 figure 2b. The violin plots look a bit confusing because all the segments are very scattered. I suggest to plot instead a smoothed curve (e.g. as produced by the default `geom_violin` function in R.)

Page 10 figure 4a. Why are there two 'n'?

Page 10 line 217. How did the authors determine the vertical/horizontal axes in their coordinate system? Did they use the horse to approximate the vertical direction?

Page 10 line 221. In a similar way, what was the reference used to calculate the absolute elevation? Was it a point on the horse or on the ground?

Page 12 line 270 'run' instead of 'running' might be better here

Page 13 line 304. ...[32], suggesting 'that' there are severe...

Author's Response to Decision Letter for (RSPB-2020-1521.R0)

See Appendix B.

Decision letter (RSPB-2020-1521.R1)

28-Jul-2020

Dear Dr How

I am pleased to inform you that your manuscript entitled "Zebra stripes, tabanid biting flies, and the aperture effect" has been accepted for publication in Proceedings B.

Your article has been estimated as being 8 pages long. Our Production Office will be able to confirm the exact length at proof stage.

Open Access

Paper charges

Sincerely,
Editor, Proceedings B
mailto: proceedingsb@royalsociety.org

Appendix A

Response to reviewer comments:

Associate Editor

Comments to Author:

Two expert referees have now carefully reviewed your manuscript, and both have surprisingly similar comments and suggestions. Both agree that your study is well written, topical and very interesting but unfortunately both feel that your main hypotheses are not supported by the data, and indeed that your data possibly open more questions than they answer. Essentially, both reviewers are of the opinion that due to the fact that you have not accounted for the optics of the compound eye - and its implications for aliasing and contrast in the optic flow experienced - disruption of the expected optic flow might not be the only explanation for your fly avoidance data (although it is definitely one possible explanation). Additionally both referees have difficulty with your "reveal" hypothesis, which the second referee suggests would require additional experiments, with a variety of other patterns (such as fractal patterns), to demonstrate. Unfortunately, you will likely need to spend a considerable amount of effort to address the concerns of the referees before the manuscript is ready for publication.

We thank the editor and reviewers for their insightful and constructive comments. We found them extremely useful and have, as a result, substantially revised the manuscript, , in particular dropping our 'reveal' hypothesis and so shifting the focus of the introduction and discussion to reframe hypotheses and interpretations..

Reviewer(s)' Comments to Author:

Referee: 1

Comments to the Author(s)

Review: How do zebra stripes prevent landings by tabanid biting flies? **We have rephrased the title to highlight the main thrust of our revised paper. It now reads: "Zebra stripes, tabanid biting flies, and the aperture effect"**

In this paper, the authors aim to investigate the function of zebra stripes and their apparent repellent effect on biting flies by exploring the mechanism by which high contrast stripes prevent landings by flies. They do this by covering horses with a range of high contrast patterns and recording landings and flight trajectories of approaching flies. The subject is of great general interest, the study is well-written, and the methodology is thorough. I enjoyed reading this paper but there appear to be two main conceptual weaknesses that severely limit the validity of the conclusions drawn and these are related to the hypothesis that, as the flies approach the high contrast patterns, they suddenly are 'revealed' to the flies and that this disrupts their targeting system when landing.

I do not understand the 'reveal' hypothesis, which the authors conclude, is the most likely mechanism preventing biting flies from landing on the tested high contrast patterns. This is because, I disagree that there is good optical evidence that, as the flies approach a pattern that becomes resolveable, it will be 'suddenly' visible in high contrast as the authors suggest on line 63 and throughout the text. I would expect that, rather than appearing suddenly, the stripes gradually become resolveable and increase in contrast as the flies approach, first being perceived as low contrast and then becoming higher contrast and lower spatial frequency. **This is a good point that we had not considered – thank you!** This is mainly due to optical principals such as the contrast transfer function: that is, that the contrast of the gratings that can be resolved by an optical system decrease as they get finer. I might have totally misunderstood

what the authors meant with this reveal hypothesis, in which case the underlying principals should be more clearly explained in the text. But I nonetheless find it difficult to understand how a pattern that becomes resolveable on approach will 'suddenly' appear. **As a result of this, and comments from reviewer 2, we have reconsidered and dispensed with the notion of 'reveal' in our revision. Instead we have restructured our hypotheses into those that affect flight control (both aperture effect, and aliasing) and those operating more centrally at the decision-making level. (Introduction lines 53-95 and reworked figure 1).**

I have difficulty understanding the logic behind the interpretation of how the different rug types affect the flies' flight manoeuvres. I do not follow the logic of the explanation given on line 220. Why would the high contrast patterns cause the flies to turn away? This point is not answered clearly. The flies could make faster turns around the patterned rugs than around the grey rug because the high contrast edges provide them with more reliable visual cues that they can use to better control fast flight. **We thank the reviewer for highlighting this lack of clarity. We have rephrased the interpretation of the speed result in the discussion (lines 254-263). We also now acknowledge that differences in speed may be caused by more reliable optic flow cues from the patterned rugs, but argue that this is unlikely, given how few landings were observed on checked rugs.**

The authors should be careful when interpreting the effect of the grey rug, as it could potentially provide the flies with just as salient optic flow cues due to contrast gain enhancements (Harris et al. 2000).

Harris, R. A., O'Carroll, D. C. & Laughlin, S. B. (2000) Contrast gain reduction in fly motion adaptation. *Neuron* 28:595-606.

Thank you for highlighting this work. We now include this in a sentence of explanation on lines 61-63.

I do not understand the reasoning given on line 225. As the flies approach a horizontal stripe, they will still experience expansion cues at the edges. If they balanced these expansion cues, they could still make effective 'optic flow'-based controlled landings. **We now explicitly detail the reasoning behind this prediction in our introduction (lines 53-81). Briefly, we argue that radially-symmetrical optic flow patterns could be necessary for guiding landing, as these would typically be experienced by flying insects when landing on a surface. Laterally dominated optic flow patterns (such as those generated by wide-field stripe patterns) may instead stimulate optic flow sensing neurons sensitive to translational flight, and so wide field stripe patterns may cause a behavioural switch in the flies from 'landing' to 'translating'. We have now specified this prediction in the introduction.**

To summarise, the logic behind the two main proposed hypotheses is not well supported in the text. As such, I do not find that the interpretations are supported by the results. To address this issue, the authors need to explain the mechanisms that they are testing more clearly and from the basic principles of vision. **We hope we have done a better job this time around by restructuring our hypotheses, and introducing and discussing them in more detail.**

Referee: 2

Comments to the Author(s)

GENERAL

The functional role of the black-and-white stripes on zebra coats has been a longstanding enigma. This

interesting study builds on a number of earlier studies that have examined the visual effect(s) that the stripes may have in deterring parasitic flies, such as tsetse flies and tabanids, from landing on the coat. A number of similar earlier studies (cited in the present manuscript, some of which come from the same laboratory) have already provided clear evidence that these flies avoid striped coats, and prefer to land on coats that do not carry stripes. The present study confirms these earlier findings - based on detailed analysis of flight speeds, approach distances and turning rates **Yes we are happy that it confirms earlier research further bolstering the case for stripes being anti-parasite devices** - and takes one further (small) step by examining the reaction to checkerboard patterns. It is found that checkerboard patterns have the same repelling effect as stripes.

Yes, it may seem like a small step, but our findings are key for discounting one of the most favoured hypothetical mechanisms for how stripe patterns operate on the visual system of horseflies, one which was outlined in several previous manuscripts, and discussed at length by visual ecologists over the last few years. We have attempted to make this case clearer in the manuscript (e.g. lines 108-120) and hope that you will agree that the finding is important and a significant advance on the long-standing conundrum.

The authors interpret this finding to mean that the deterrent effect of stripes is not due to “disruption” of optic flow (caused by an aperture effect), but to what they describe as a phenomenon of “revelation”. It is true that stripes generate an unusual pattern of optic flow that is different from what would normally be encountered by a landing insect. However, I do not see a strong a priori reason to suspect that this may be the cause - given that many insects, including bees, show no tendency to avoid landing on striped patterns. **We have combed the literature in an attempt to find examples of work in which flying insects are tasked with landing on wide-field stripe patterns but have had no success. There are many examples using localized grating patterns, for example to test learning and resolution in honeybee training experiments (e.g. Horridge 2003, JEB 206: 2105-2110), or to show that local low and high contrasting edges affect landing speed (e.g. Baird et al 2013 PNAS 110:18686-91). But none that we know present flying insects with wide-field horse-sized stimuli. The smaller stimuli tested so far in the literature offer plenty of veridical optic flow cues from the edges of patterns and the surrounding environment. We think that this would be a worthwhile avenue of research and preliminary experiments by M How using trained bumblebees suggests that they do indeed have an aversion to landing on inconspicuous targets located near the surface of wide-field stripe patterns (compared to other high contrast patterns).**

While the disruption hypothesis is still plausible, I am not sure that the experimental findings exclude other possibilities. For example, any periodic pattern – be it a linear grating or a checkerboard – would be subject to spatial aliasing when it is viewed by the insect’s compound eye. When the (angular) period of the stripes (λ) is larger than twice the interommatidial angle ($\Delta\phi$), the stripes (or checkerboard) would appear to move in the same direction in the insect’s eye as they do in the real world. However, when λ is smaller than twice $\Delta\phi$ (which will occur when the insect is far away), the stripes (or checkerboard) would appear to move in the opposite direction. This effect of spatial aliasing in the fly’s eye was demonstrated by Goetz and Reichardt back in the sixties. **Thank you for highlighting this. We agree that this is an important alternative hypothesis. The senior author has already explored in this possibility in an earlier paper (How & Zanker, *Zoology* 2014).** When the fly is very far away from the pattern, the pattern will appear grey because its (angular) spatial frequency is higher than the cutoff angular spatial frequency of the modulation transfer function (MTF) of the visual system, which is determined by the width ($\Delta\rho$) of the angular acceptance function. (The optics functions as a spatial low-pass filter). As the fly approaches the pattern the angular period of the pattern will progressively increase, and the apparent contrast of the pattern will increase to the point where the pattern becomes visible (i.e. the

contrast is suprathreshold). If the angular period of the pattern at this distance is smaller than twice the interommatidial angle, then spatial aliasing would occur and the apparent movement of the pattern would be in a direction opposite to that which would occur in a normal visual environment - which is dominated by low, non-aliasing spatial frequencies. This contradictory movement would work against the tendency for the fly to fixate the pattern, creating positive feedback and causing it to turn away from the pattern. **Thank you for explaining this process so clearly!** For the stripe and checkerboard patterns used in these experiments, the spatial period is 7.0 cm, which, at a viewing distance of 70 cm, would correspond to an angular period of 0.1 radians, or about 5.7 degrees. Thus, spatial aliasing would occur at this distance if the interommatidial angle is larger than half this value (2.85 degrees). This value of interommatidial angle is not unrealistic for flies, and the distance of 70 cm for fast turns corresponds approximately to the values shown in Fig 5d. That is not to say that spatial aliasing is the correct explanation for the findings – it is just an additional possibility that the authors may wish to consider. **We agree and as a result of this comment and points raised by reviewer 1, we have restructured our hypotheses to incorporate this more clearly. We hope you don't mind, but we also borrowed parts of your clear explanation and modified it for incorporation in our text. Thank you.**

I find it difficult to understand the 'reveal' hypothesis that is offered as an explanation for the present findings. I can see how the stripes and the checkerboard patterns - which cannot be resolved at large distances and will therefore appear grey - will suddenly be 'revealed' to the insect when it comes closer. But what about coats that carry random spots, blobs or patches, which also seem to repel the flies (see, for example, Blaho et al. 2012)? Such patterns would be at least partially visible at all distances – there would be no sudden 'revelation', as far as I can see. To a fly's poor eyesight, even large blocks of black on white might appear grey at a distance. With the textureless coats, there would have to be some (minimal) contrast on the coat, to provide the optic flow that flies rely upon for executing smooth landings. **Correct – now included in the Introduction!** Now, if we are saying that there is no sudden 'revelation' in these 'textureless' coats, does this imply that the low-contrast textures in such coats have the same appearance and spatial properties at all scales of magnification – rather like a fractal pattern, or a scene with a power spectrum of $[1/(\text{spatial frequency})^2]$? If that is what the authors are saying, then, according to the 'reveal' hypothesis, the flies should show no hesitation towards landing on a high-contrast fractal pattern. This prediction could be tested quite easily. **Once again, we thank the reviewer for making us reconsider the 'reveal' hypothesis and for providing good reasons to dispense with it. We hope that our newly restructured hypotheses will help to reframe the questions more logically. Also, thank you for providing ideas about fractal patterning, which we are now endeavouring to tackle in the 2020 field season.**

As far as I can see, this study does indeed thoroughly exclude one possibility – namely, that the avoidance of stripes on a zebra is not due to the disruption of the expected optic flow pattern, caused by the aperture effect. **Yes, we agree and have now refocused the paper on this main finding.** But in doing so, the results and their interpretation raise more questions, and do not seem to provide an answer for the reason why these flies avoid landing on an animals' coat whenever it carries *any* pattern – be it stripes, checkerboards, or spots or patches. The 'appearance' explanation therefore does not make sense to me, because it does not seem to provide a logically consistent explanation for all of the observed behaviors that have been documented in the literature. It seems to me that this calls for more experimental work using a variety of different patterns.

Having thought about this comment long and hard we agree that the data in this paper can only go as far as excluding the aperture effect as a possible mechanism. However, the aperture effect continues

to be held up as the number one explanation for the underlying cause of why horseflies avoid zebra stripes, and so we think our results are important in disproving this. The consistency with which our five sets of results point away from the aperture effect moves the field forward in an important way and clearly sets out predictions for future experimental work.

SPECIFIC

Lines 80- 83: This statement would not be true if the grey rug has zero contrast, because it would not generate any optic flow. Need to specify the properties that you assume a 'grey' rug to have. Same issue with the statement in lines 86-88. **We have included a statement highlighting that the flies almost certainly make use of contrast gain adjustment for optic flow motion sensing so that the low-contrast features on the grey rug will likely generate sufficient optic flow for flight guidance (lines 61-63).**

Lines 99-100: I presume the flies are not marked, so they cannot be identified individually. Can you provide some idea of how many flies were likely to be participating in the experiments, so that one can get a feel for the statistical robustness of the data? Landings from multiple individuals would be preferable to multiple landings from a single fly, because the former would enhance statistical independence. **While we cannot rule out that the occasional trajectory came from the same individual, we minimized this risk in the tracking data by sampling broadly across time. We now include a sentence of explanation on lines 151-153: "Care was taken to sample the 100 or so trajectories from each rug type across as many horses as possible, over as wide time intervals as possible to further reduce the unlikely event of digitizing the same fly".**

Legend to Figure 3: When you say $n=107$ flies or $n=101-107$ flies per rug type, I presume you the number of flights analyzed, and not the number of individual flies. **Correct, now made explicit.**

Figure 3a: Might be useful to show trajectories for other patterns too, including the grey rug, in the Supplementary Information section. **Supplementary figure and video now included for all trajectories on all rugs.**

Figure 3b: Not clear what you mean by 'stacked histogram' here. This term usually applies to histograms in which bars are stacked vertically on top of each other, with each bar representing the count of a different variable within the same bin. That does not seem to be the case here, and it is unclear what the bars mean. Is it simply a histogram showing bin counts at various distances? Same thing in Fig. 4b: Bin counts at various speeds? Such plots are usually termed 'violin plots'. **Corrected to 'violin plots'.**

Fig. 5b and legend: Is the flight segmented into fast and slow speeds, or fast and slow turning rates? Please clarify. The colors seem to correspond to those used in the speed histogram shown in Fig. 5a, and not to those used in the yaw and pitch histograms. **Yes, this was tricky to show clearly. The flight trajectory is separated into both speed AND turn rate. The fast turns correspond to flight segments that move fast (blue area in speed histogram) and turn (pink areas in yaw and/or pitch histograms). Slow turns are the same, but for slow segments (red area under speed histogram). We continued the red/blue labelling in b for fast/slow. Grey points are all the segments falling within the grey area of the yaw and pitch histograms.**

Lines 223-224, where you say "an optic flow hypothesis would predict any turns would occur at similar distance for grey and checked rugs": This would be true only if (a) the grey rugs display at least a low

contrast and (b) the movement sensing neurons in the fly's visual system signal optic flow (image speed) veridically, independently of the contrast or texture of the image. **We have now modified this point to specifically address the aperture effect hypothesis.**

Appendix B

Referee: 2

Thank you for these revisions.

I am glad to note that the manuscript has now been revised to remove the previous conclusion about the role of the 'appearance' hypothesis, and to now conclude that the avoidance of stripes is not caused by the aperture effect. The real reason for the increased avoidance of coats decorated with stripes and checkerboard patterns (as opposed to grey and black coats) remains to be investigated, but at least the revised conclusion of the study is now compatible with the data.

In lines 117-119: I think it would be crucial to add here that, if the aperture effect were responsible for landing avoidance, then you would expect avoidance only for the striped patterns, and not for the checkerboard, grey or black patterns.

- Thank you for the suggestion. We added the sentence: "If the aperture effect were responsible for landing avoidance, then the effect would be expected only for the striped patterns, and not the checked, grey or black patterns" to lines 120-122.

Referee: 1

I appreciate the changes that the authors have made to address the concerns of the reviewers. In general, I am satisfied with the response. I just have one remaining comment and a few suggestions for changes.

Page 10 line 196. The criteria used to determine slow/fast turns should be defined in the method section. More generally, what is the biological relevance of this categorization? Without further justification, calculating these parameters seems a bit artificial. How do the specific values used for categorization (fig 5a) affect the results (figure 5c-d)?

- Thank you for highlighting this. We have incorporated details of how and why these values were chosen to represent speed and turn categories in the methods section lines 159-165: "Digitised segments were divided for further analysis into slow/fast flight and straight/turning flight by determining whether pairs of trajectory points fell below (slow) or above (fast) the 65th percentile speed value of 2.35 m/s and whether triplets of points showed an angular turn rate of above or below the 65th percentile value of 390°/s in azimuth or 210°/s in elevation. These thresholds were chosen arbitrarily, based on the shape of histograms of the complete dataset (see results), but it must be noted that percentile thresholds between 50 and 70 all produce similar results."

Finally, I am surprised that some of the first segments of the example trajectory given in figure 5b could be considered as turns as the start of the trajectory seems relatively straight.

- This is because some of the turn behaviour is occurring along the plane of view, and so the curve is not as visible when viewed in two dimensions. We have attempted to remedy this by rotating the plot to a more optimum position in the figure.

object is approached is not always true (for example if the animal continuously decreases speed). I suggest specifying that the fly approaches 'at a constant speed'.

- This comment was truncated, but we think refers to line 57. This has now been amended as suggested: “As a fly moving at a constant speed approaches an object, the surface looms ever faster...” (line 57-58).

Page 3 line 91. Away from which point?

- Reworded to clarify – between approximately 0.5-1m from the horse. Lines 92-93

Page 5 line 143-148. Was any noise-removal or smoothing procedure involved when reconstructing the fly trajectories? If yes, this should be specified in the methods.

- Thank you very much for spotting this omission. Yes, the data were smoothed with a three-point moving average. This is now specified in lines 153-154.

Page 7 line 148-149. Did the authors consider movements of the horses during or between recordings?

- Yes, trajectories were only digitised around stationary horses. This is now clarified in line 151.

Page 8 figure 2b. The violin plots look a bit confusing because all the segments are very scattered. I suggest to plot instead a smoothed curve (e.g. as produced by the default `geom_violin` function in R.)

- Thank you for the suggestion. We are reluctant to use smoothed violin plots as, in this case, it will give the illusion of more data than there is. We feel that the fractured blocky nature of the violin plot gives an idea of the data distribution while acknowledging that this is based on an n of only 22 per treatment. The key data is contained in the median and quartile lines. As a result, we are keen to preserve the figure as it is. Alternatively, if this is not acceptable, we could simplify the plot by removing the violin plot data, but this will then lose interesting information.

Page 10 figure 4a. Why are there two ‘n’?

- The n-number was different for each of the four treatments, ranging from 101 to 107. This is also the case for figs 3b-c, 5, and 6. This is now clarified at the end of the methods (lines 165-166)

Page 10 line 217. How did the authors determine the vertical/horizontal axes in their coordinate system? Did they use the horse to approximate the vertical direction?

- The stereo camera rig was positioned to approximately maintain horizontal/vertical axes with the outside world. This is now clarified in the methods section lines 142-143

Page 10 line 221. In a similar way, what was the reference used to calculate the absolute elevation? Was it a point on the horse or on the ground?

- Absolute elevation was calculated relative to the horizontal/vertical axes of the camera system, which in-turn was positioned to approximate the horizontal/vertical axes of the real world.

Page 12 line 270 'run' instead of 'running' might be better here

- Here we disagree. We think 'running' is the correct grammar in this case.

Page 13 line 304. ...[32], suggesting 'that' there are severe...

- Amended as suggested – line 317.